# The red flags of ulnar neuropathy in leprosy

**Márcia Jardim** [1,2,3]* *, **Robson T. Vital** [2], **Ximena Illarramendi** [1], **Mariana Hacker** [1],
**Beatriz Junqueira** [1], **Izabela J. R. Pitta** [1,2,3], **Roberta O. Pinheiro** [1], **Euzenir N. Sarno** [1]

**1** Oswaldo Cruz Institute, Fiocruz, Rio de Janeiro, Brazil, **2** Department of Neurology, Pedro Ernesto
University Hospital/Rio de Janeiro State University, Rio de Janeiro, Brazil, **3** Federal University of the State of
Rio de Janeiro, Rio de Janeiro, Brazil

☯ These authors contributed equally to this work.
* Jardim.mm@gmail.com

## Abstract

The diagnosis of pure neural leprosy is more challenging because patients share characteristics with other common pathologies, such as ulnar compression, which should be taken into consideration for differential diagnosis. In this study, we identify ulnar nerve conduction characteristics to aid in the differential diagnosis of ulnar neuropathy (UN) in leprosy and that of non-leprosy etiology. In addition, we include putative markers to better understand the inflammatory process that may occur in the nerve. Data were extracted from a database of people affected by leprosy (leprosy group) diagnosed with UN at leprosy diagnosis. A non-leprosy group of patients diagnosed with mechanical neuropathy (compressive, traumatic) was also included. Both groups were submitted to clinical, neurological, neurophysiological and immunological studies. Nerve enlargement and sensory impairment were significantly higher in leprosy patients than in patients with compressive UN. Bilateral impairment was significantly higher in the leprosy group than in the non-leprosy group. Leprosy reactions were associated to focal demyelinating lesions at the elbow and to temporal dispersion (TD). Clinical signs such as sensory impairment, nerve enlargement and bilateral ulnar nerve injury associated with eletrodiagnostic criteria such as demyelinating finds, specifically temporal dispersion, could be tools to help us decided on the best conduct in patients with elbow ulnar neuropathy and specifically decide if we should perform a nerve biopsy for diagnosis of pure neural leprosy.

## Introduction

Every person affected by leprosy suffers, to some degree or another, from peripheral nerve involvement as a result of the disease. This may range from the alteration of intradermal nerves in a patch of skin to a major lesion on a nerve trunk. Fortunately, neuropathy, a feared and challenging complication of leprosy, is treatable if timely diagnosed. In patients with pure neural leprosy, in which there are no associated skin lesions, diagnosis is even more challenging because of other common pathologies considered in the differential diagnosis, especially if it exclusively affects the ulnar nerve [1]. Involvement of the ulnar elbow segment is one of most common causes of focal neuropathy of the upper limbs, second only to the median nerve

**Competing interests:** No authors have competing interests.

compression of the wrist segment in carpal tunnel syndrome [2]. Ulnar neuropathy (UN) at the elbow is one of the most prevalent in leprosy at any stage or clinical presentation of the disease [3].

Altered patterns of nerve conduction together with clinical criteria are useful for identifying the cause of UN [4]. Several authors specifically and deeply assess the function of this nerve [5] but rarely include leprosy [6, 7]. Aside from neurophysiological signs of leprosy neuropathy, additional markers such as the inflammatory cytokines tumor necrosis factor (TNF) and interleukin 6 (IL-6), that are produced by residual and infiltrating macrophages, mast cells, lymphocytes, Schwann cells, fibroblasts, and sensory neurons, could also be useful in establishing the cause of UN [8]. High levels of TNF in neurological disorders have been associated with demyelination, axonal degeneration, increased nerve blood barrier permeability and immune recruitment to the site of injury [9]. On the other hand, IL-6 production has been associated with proinflammatory signaling in Schwann cells. Leprosy reactions are a common complication of the disease and may occur before, during, or after release from treatment. During these reactions, a high level of inflammatory mediators are produced by the reactivation of the cellular immune response and may lead to substantial nerve damage [10–12]. TNF, in particular, is known to induce IL-6 and IL-8 production by Schwann cells [10]. Leprosy neuropathy is mainly a multiple mononeuropathy with axonal loss, but it is rarely recognized as a cause of demyelination or conduction block. In this study, we identify ulnar nerve conduction characteristics to aid the differential diagnosis of UN from non-leprosy etiology. In addition, we include putative markers to better understand the inflammatory process that may occur in the nerve.

## Methods

A cross-sectional observational study was performed with patients diagnosed at the Souza Araújo Outpatient Clinic (ASA), Oswaldo Cruz Institute, Fiocruz, a national public health system reference center for leprosy, between April 2009 and April 2013. Data were extracted from the nerve conduction studies (NCS) database from people affected by leprosy (leprosy group) diagnosed with UN at leprosy diagnosis, during and/or after multidrug therapy (MDT). A non-Leprosy group of patients diagnosed with mechanical neuropathy (compressive, traumatic) after suspicion of having the pure neural form of leprosy were also included. Patients received fixed-dose paucibacillary (PB) or multibacillary (MB) MDT according to the results of their slit-skin smears or skin/nerve biopsies. The research was carried out in compliance with the International Norms on Ethics in Human Research, having been previously approved by the Ethics Committee of the Oswaldo Cruz Foundation (Approval number: 21625313.9.0000.5248). All patients voluntarily provided their written informed consent.

The events of leprosy reactions were recorded and classified as follows: (1) reversal reactions (RR) involve inflammation of preexisting lesions with tender, new lesions and/or acroedema; (2) *erythema nodosum leprosum* (ENL) is the sudden appearance of inflamed papules, nodules, and plaques, with or without systemic signs and symptoms (fever, malaise, arthralgia, lymphadenopathy, etc), and (3) neuritis when one or more enlarged nerves is painful or presents loss of function. Neuritis may be isolated or arise together with cutaneous and systemic signs and symptoms.

### Neurological examination

We performed a detailed neurological examination to record the number and distribution of affected peripheral nerves. The assessment included evaluation of sensory impairment (tactile sensation using the monofilament test, mechanical nociception, and cold

thermosensation), and motor function by the voluntary muscle test as described in detailed by Vital *et al.* (2012) [13].

## Nerve conduction study

Electrophysiological testing was performed using the Neuropack 2 system (Nihon Kohden Corp., Tokyo, Japan). Room temperature was maintained between 29-32˚C. Patients were placed in a supine position. Skin temperature at the wrists was kept above 33˚C with the use of ultrared light, when necessary. Nerve conduction data was collected using a band pass filter setting of 2 Hz-10 KHz.

Sensory NCS (sNCS): The ulnar sensory nerve action potentials (SNAPs) were orthodromically recorded at a fixed recording and stimulation distance of 12 cm.

Motor NCS (mNCS): The ulnar nerve was stimulated at the wrist, below and above the sulcus (on the flexed elbow), and at the arm and was recorded at the abductor *digiti minimi muscle* [14]. The amplitudes of proximal and distal Compound Muscle Action Potentials (CMAPs) were compared across the different stimulation sites: from below the elbow to the wrist (forearm), from above to below the elbow (elbow), and above the elbow (arm). Normal cut-off values were defined according to Fuglsang-Frederiksen, 2010 (Box 1) [15].

### Box 1. Normal cut-off values for ulnar nerve conduction study

| Ulnar nerve function | Parameter | | |
|---|---|---|---|
| | amplitude | velocity | latency |
| Sensory | 8 μV | 43m/s | 2.7 ms |
| Motor | 4mV | 55m/s | 3.3 ms |

Temporal dispersion (TD) was defined as a reduction in proximal CMAP amplitude compared with distal CMAP amplitude when the proximal CMAP duration increases by > 30% and/or a > 15% in area [16]. Conduction block (CB) was defined by a reduction of proximal CMAP area/amplitude by > 50% compared with distal CMAP area/amplitude. The duration of the proximal CMAP should not increase by > 30%.

The UN was classified according to CMAPs and SNAPs parameter values: a) no neuropathy, if normal values were registered (box); b) axonal involvement, defined by a CMAP and/or SNAP reduction, amplitude reduction of < 30% of reference values and the sensory and/or MVC above 70% of the reference value; c) demyelinating impairment defined by prolonged latency of CMAP and/or SNAP compared to the reference value, together with a reduction of sensory and/or motor velocities to < 85% of the reference value; d) mixed, if there were axonal and demyelinating lesions. If SNAPs and CMAPs could not be obtained by the above mentioned method, the nerve was classified as "no conduction" and excluded from the analysis to describe the pathophysiological process.

## Serum levels of inflammatory mediators

Sixteen serum samples from patients affected by leprosy UN were selected, as described in the flowchart. The selection criteria was the presence of CB. The influence of reactional episodes or neuritis was also analyzed within groups with or without CB. Blood samples from leprosy patients were previously collected and processed under sterile conditions. Sera were extracted

and stored at -20˚C until use, according to biosafety and good laboratory practices. Concentrations of proinflammatory mediators TNF and IL-6 were measured in the sera by commercial enzyme-linked immunosorbent assay (ELISA) following the manufacturer instructions (eBioscience- San Diego, CA, USA). Cytokine levels were measured in picograms per mL (pg/mL) and each value represents the mean of duplicate samples.

## Statistical analysis

Descriptive statistics (frequencies, proportions and mean) were performed for both groups. Groups were compared using the chi-square test ($\chi^2$). The level of significance established was 5% and two-sided p-value considered. Cytokine levels were compared using an unpaired t test in GraphPad Prism version 5 (GraphPad Software).

## Results

Mean age of patients affected by leprosy (48.5 years) was similar to that of non-leprosy patients (49.3 years of age), although in the leprosy group there was a higher percentage of patients from 50 to 65 years of age (Table 1).

## Clinical findings

Forty seven percent of patients in the leprosy group did not present any disability at the beginning of MDT and clinical classifications were evenly distributed according to treatment. Twenty five percent of patients presented neuritis (either isolated or associated with reversal reaction) at the beginning of treatment.

**Ulnar nerve alterations.** Nerve enlargement and sensory impairment were significantly higher in leprosy patients (0,001 and 0,003, respectively) than in patients with compressive UN (Table 2).

## Neurophysiological findings

**Sensory conduction study.** When evaluating sensory nerve conduction classification, more cases presented no conduction in the leprosy group than in the non-leprosy group (p = 0.001). On the other hand, patients in the non-leprosy group presented a higher proportion of axonal lesions than those in the leprosy group. During reactional episodes we observed

**Table 1. Socio-demographic characteristics and clinical ulnar alterations observed in leprosy patients and people with compressive ulnar neuropathy.**

| | | | Leprosy (n = 93) | Non leprosy (n = 36) | $\chi^2(p)$ |
|---|---|---|---|---|---|
| Age (years) | | | (12–90) | 49.3 (13–83) | |
| | Age range | 12–34 | 26 (28%) | 8 (22%) | |
| | | 35–49 | 18 (19%) | 12 (33%) | |
| | | 50–65 | 34 (37%) | 11 (31%) | |
| | | >66 | 15 (16%) | 5 (14%) | |
| Gender | | Female | 28 (30%) | 18 (50%) | |
| | | Male | 65 (70%) | 18 (50%) | |
| Ulnar nerve alterations (signs and symptoms) | | Paresthesia | 67 (72%) | 26 (72%) | $\chi^2$ (0,184) |
| | | Neural pain | 34 (37%) | 8 (22%) | $\chi^2$ (0,059) |
| | | Nerve enlargement | 54 (58%) | 10 (28%) | $\chi^2$ **(0,001)** |
| | | Sensory impairment | 81 (87%) | 23 (64%) | $\chi^2$ **(0,003)** |
| | | Motor impairment | 50 (54%) | 22 (61%) | $\chi^2$ (0,451) |

**Table 2. Clinical characteristics of 93 persons affected by leprosy with ulnar neuropathy.**

| Disability grade (according to the World Health Organization) | | 0 (no disability) | 25 (47%) |
|---|---|---|---|
| | | 1 | 15 (28%) |
| | | 2 | 13 (25%) |
| | | Not available | 40 |
| Clinical forms | | LL | 18 (19%) |
| | | BL | 11 (12%) |
| | | BB | 8 (9%) |
| | | BT | 21 (23%) |
| | | TT | 1 (1%) |
| | | I | 9 (10%) |
| | | PN | 19(20%) |
| | ND | | 6 (6%) |
| Treatment | | PB | 50 (54%) |
| | | MB | 43(46%) |
| Reaction at diagnosis | | No reaction | 54 (58%) |
| | | RR | 11 (12%) |
| | | ENL | 5 (5%) |
| | | Neuritis | 13 (14%) |
| | | RR+Neuritis | 10 (11%) |

PB = Paucibacillary; MB = Multibacillary; LL = Lepromatous; BL = Borderline lepromatous; BB = Borderline borderline; BT = Borderline tuberculoid, TT = Tuberculoid, I = Indeterminate, PN = Pure Neural, RR = Reversal Reaction; ENL/EM = , *erythema nodosum leprosum*/erythema multiform.

a small increase in nerve damage, however, there was no difference in grade of damage (p = 0.636) (Table 3).

**Motor conduction study.** Motor function was normal for most of the nerves in both groups of patients, 107 (58%) and 46 (62%) in the leprosy and non-leprosy groups, respectively. Bilateral impairment was significantly higher in the leprosy group than in the non-leprosy group (Table 4). There was no difference in the distribution of altered nerves regarding the classification of the type of lesion (ie, pathophysiological) by the motor NCS between the groups, even after exclusion of non-classified and no conduction nerves. Both groups show demyelinating predominance. The frequency of TD in the Leprosy group was significantly

**Table 3. Sensory nerve conduction study findings (classification of sensory ulnar nerves according pathophysiological pattern).**

| | | | No reaction (n = 108) | Leprosy (n = 186) | | | | | Total (n = 186) | Non Leprosy (n = 72) |
|---|---|---|---|---|---|---|---|---|---|---|
| | | | | Reaction (n = 78) | | | | | | |
| | | | | RR (n = 22) | RR+ Neuritis (n = 20) | Neuritis (n = 26) | ENL (n = 10) | Total (n = 78) | | |
| Classification Nerve Conduction study n (%) | Sensory | Normal | 44 (40%) | 5 (22%) | 3 (15%) | 10 (38%) | 3 (30%) | 21 (27%) | 65 (35%) | 35 (49%) |
| | | Axonal Lesion | 8 (7%) | 3 (14%) | 5 (25%) | 2 (8%) | 2 (20%) | 12 (15%) | 20 (11%) | 9 (13%) |
| | | No conduction | 45 (41%) | 10 (45%) | 7 (35%) | 12 (46%) | 4 (40%) | 33 (42%) | 78 (42%) | 12 (17%) |
| | | No classification | 11 (10%) | 4 (18%) | 5 (25%) | 2 (8%) | 1 (10%) | 12 (15%) | 23 (12%) | 16 (22%) |

n = number of sensory ulnar nerves; RR = Reversal Reaction; ENL = , erythema nodosum leprosum.

**Table 4. Motor nerve conduction study findings.**

| mNCS | | Leprosy group | Non leprosy group | p-value |
|---|---|---|---|---|
| | Unilateral alteration | 27 (14%) | 26 (35%) | 0.000001 |
| | Bilateral alteration | 52 (28%) | 2 (3%) | |
| Total | | 79 (100%) | 24 (100%) | |
| Pattern of ulnar nerve lesion | Dem | 34 (43%) | 10 (36%) | 0.7982 |
| | Axonal | 14 (18%) | 8 (29%) | |
| | Mixed | 14 (18%) | 5 (18%) | |
| | NC | 8 (10%) | 2 (7%) | |
| | No Classification | 9 (11%) | 3 (10%) | |
| Total of altered mNCS | | 79 (100%) | 28(100%) | |
| CB* | No | 170 (96%) | 66(92%) | P = 0.1162 |
| | Yes | 8 (4%) | 6 (8%) | $X^2 = 1.429$ |
| TD* | No | 148 (83%) | 66(92%) | *P = 0.041* |
| | Yes | 30 (17%) | 6 (8%) | $X^2 = 3.019$ |
| CB* ulnar nerve segment | Forearm | 2 (25%) | 1 (17%) | P = 0.6154 |
| | Elbow | 6 (75%) | 5 (83%) | |
| | Arm | 0 (0%) | 0 (0%) | |
| TD* ulnar nerve segment | Forearm | 17 (57%) | 3 (50%) | P = 0.1979 |
| | Elbow | 6 (20%) | 3 (50%) | |
| | Arm | 7 (23%) | 0 (0%) | |

mNCS = motor nerve conduction study; Dem = demyelinization; NC = No Conduction; CB = conduction block; TD = temporal dispersion;

* exams were excluded when there was no conduction in the ulnar nerve (8 nerves in the leprosy group and 2 nerves in the non leprosy group).

higher than CB (CB x TD = p = 0.000079 X2 = 14.26). In the non leprosy group the frequency of TD was equal to that of CB. There was no difference when comparing CB in the leprosy group with CB in the compressive group together at any point. Exams of nerves with no conduction were excluded (Table 4).

Leprosy reactions were associated to focal demyelinating lesions at the elbow and TD (Table 5). However, there was no statistically significant difference in the distribution of CB and TD in the different reaction types (TD, p = 0.1561); (CB, p = 0.6794) (Table 6). In addition, there was no statistically significant difference in the distribution of the occurrence of CB and TD in the groups with neuritis and that without neuritis (TD, p = 0.3510); (CB, p = 0.2876) (Table 7).

The type of UN was significantly different between patients with or without leprosy reactions (*P = 0.00006*, $X^2 = 24.44$) (Table 8).

**Table 5. Evaluation of type of segmental demyelination in the leprosy group with any leprosy reaction.**

| | | Leprosy reaction | | |
|---|---|---|---|---|
| | | No (n = 102)* | Yes (n = 76) * | p value |
| conduction block | No | 98 (96%) | 72 (95%) | 0.341 |
| | Yes | 4 (4%) | 4 (5%) | |
| temporal dispersion | No | 91(89%) | 57(75%) | *0.006* |
| | Yes | 11(11%) | 19(25%) | $X^2 = 6.28$ |

* nerves with no conduction were excluded (6 nerves in non-reaction leprosy patients and 2 nerves in leprosy reaction patients).

**Table 6. Evaluation of type of segmental demyelination by types of leprosy reaction.**

| | | | Reactions* | | | | |
|---|---|---|---|---|---|---|---|
| | | | RR (n = 21) | ENL (n = 10) | Neuritis (n = 25) | RR+neuritis (n = 20) | P value |
| conduction block | No | | 20 (95%) | 10 (100%) | 23 (92%) | 19 (95%) | p = 0.6794 |
| | Yes | | 1 (5%) | 0 | 2 (8%) | 1 (5%) | |
| temporal dispersion | No | | 16 (76%) | 8 (80%) | 18 (72%) | 15 (75%) | p = 0.1561 |
| | Yes | | 5 (24%) | 2(20%) | 7 (28%) | 5 (25%) | |

RR = Reversal Reaction; ENL = *Eritema nodosum leprosum*;

*excluding those with no nerve conduction (6 nerves in non-reaction leprosy patients and 2 nerves in leprosy reaction patients—1 with neuritis and 1 with RR).

## Cytokine levels

TNF and IL-6 levels were evaluated in serum samples from people affected by leprosy and UN with or without ulnar CB. As observed in Fig 1, IL-6 levels were significantly higher in sera from patients with CB when compared with samples from patients without CB (p = 0.002). On the other hand, there were no significant differences for TNF levels (p-value = 0.16).

To ascertain if the increased levels of IL-6 would be derived from reactional episodes or neuritis we selected the serum from leprosy patients without reactional episodes or neuritis and evaluated IL-6 and TNF levels by ELISA (Fig 2). Surprisingly, IL-6 levels were still statistically different between the evaluated groups. Together, these data suggest that IL-6 could be involved CB, a specific nerve impairment that is not related either to the cutaneous reactional episodes or neuritis.

## Discussion

Leprosy is an example of an infectious disease that affects the peripheral nervous system (PNS). The diagnosis of leprosy is often delayed, mainly when there is neurological impairment without the classic hypo/anesthetic dermatological lesions, such as in pure neural leprosy. Neuritis is the classic clinical picture of leprosy and is featured by thickening and pain on palpation and nerve damage associated with sensory or sensory and motor impairment [17]. The non-toxic nature of *M. leprae* guarantees the survival of Schwann Cells (SC) during long-term incubation without inducing apoptosis or toxicity. *M. leprae* is likely to reside and replicate in SC for a long time before immune cells actively enter the scene and initiate an inflammatory response, which eventually manifests clinically as this classic neuritis [18].

The ulnar nerve is classically known as one of the nerves most affected by leprosy at any stage and/or clinical form of the disease in the medial epicondyle [3]. However the

**Table 7. Evaluation of type of segmental demyelination during episodes of neuritis.**

| | | | Reaction * | | |
|---|---|---|---|---|---|
| | | | No neuritis (n = 31) | Neuritis (isolated or with RR) (n = 45) | P value |
| conduction block | No | | 30 (97%) | 42 (93%) | p = 0.2876 |
| | Yes | | 1 (3%) | 3 (7%) | |
| temporal dispersion | Não | | 24 (77%) | 33 (73%) | p = 0.3510 |
| | Sim | | 7 (23%) | 12 (27%) | |

RR = reversal reaction CB = conduction block; TD = temporal dispersion excluding those with no nerve conduction (6 nerves in non-reaction leprosy patients and 2 nerves in leprosy reaction patients—1 with neuritis and 1 with RR).

**Table 8. Type of nerve lesions during reactions episodes.**

|  | Reaction | | P valor |
|---|---|---|---|
|  | No (n = 102) | Yes (n = 76) |  |
| Normal | 72 (70%) | 35 (46%) | *0.0001* |
| Axonal | 10 (10%) | 4 (5%) | $X^2 = 24.44$ |
| Demyelinating | 10 (10%) | 24 (32%) |  |
| Combined | 9 (9%) | 5 (7%) |  |
| NC | 1 (1%) | 8 (10%) |  |

NC = no conduction.

involvement of the ulnar nerve in the elbow segment is also the second most common cause of neuropathy in the upper foci, behind only to the compression of the median nerve in the wrist segment—carpal tunnel syndrome [2]. For this reason a careful investigation should be carried out for the differential diagnosis in a patient with ulnar neuropathy, especially in leprosy endemic areas.

Our clinical findings corroborate those of other authors that demonstrated that in leprosy neuropathy sensory impairment occurs before motor impairment [19, 20]. This fact may be related to the assessment of fine fibers that are affected early in leprosy [20, 21]. On the other hand, external compression of the ulnar nerve may preferentially affect the fibers within the fascicle nearest to the compression, thereby affecting the muscle that those fibers innervate [22]. The bilateral predominance of ulnar impairment in the leprosy group in relation to patients with compressive neuropathy could be explained by the fact that, in leprosy, nerve damage presents a more diffuse pattern [23]. Nerve enlargement was another parameter which was statistically significant in comparison to compressive the neuropathy group. This is a classic feature in leprosy neuropathy [1].

Electrodiagnostic criteria play a crucial role in the routine investigation of nerve conduction studies (NCS). They could provide information about myelin sheath and axons of diseased peripheral nerves [24]. The defining electrodiagnostic criteria for primary demyelination include not only the slowing of motor conduction, but also abnormal temporal dispersion of the CMAP and motor conduction block. Ulnar-nerve entrapment at the elbow is diagnosed when there are demyelinating finds in the NCS in the elbow segment of the nerve [25]. We did

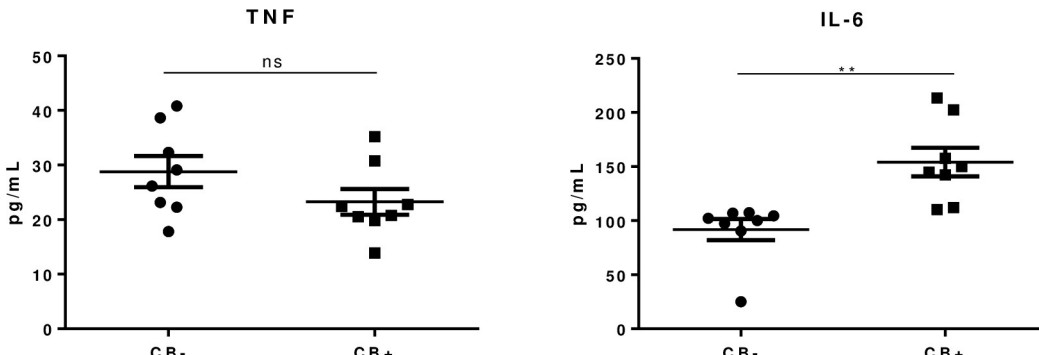

**Fig 1. Serum levels of TNF and IL-6 in patients affected by leprosy with ulnar neuropathy and conduction block (CB+, n = 8) and without conduction block (CB-, n = 8).** Patients with leprosy neuropathy and CB+ presented significantly increased levels of IL-6, in comparison with CB- patients (P = 0.002). Unpaired t test was performed. TNF levels were similar for both groups.

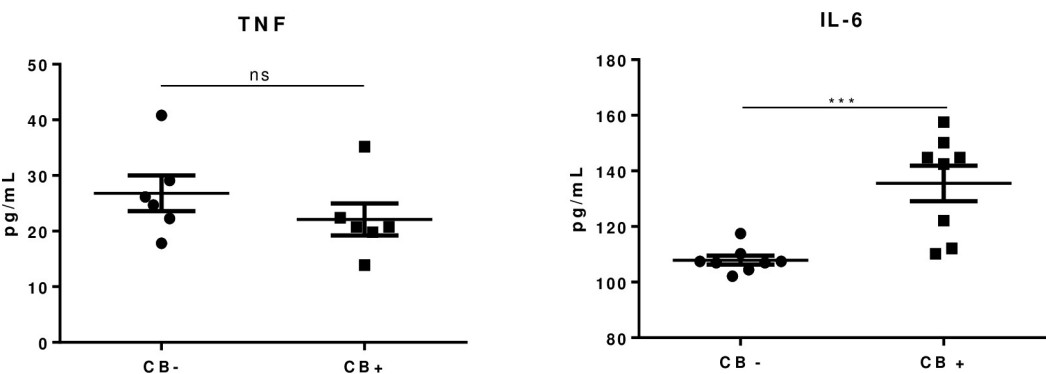

**Fig 2. Serum levels of TNF and IL-6 in leprosy patients with conduction block (CB+) and without conduction block (CB-) that did not have reactional episodes or leprosy neuritis (n = 6).** For IL-6, CB+ was significantly higher in comparison with CB- (P = 0.0009). TNF levels were not statistically different (P = 0.30). Unpaired t test was performed.

not find statistical significance in relation to CB between the leprosy and non-leprosy groups. It could be because most compressive neuropathy did not meet the criteria for CB identification.

We found more demyelinating features in both groups, however, in the leprosy group, TD was more frequent and statistically significant. This could be explained due to the physio-pathology of leprosy neuropathy. The asymmetry of fiber impairment among the fascicles is a feature of this pathology, especially during reactions when there is rapid demyelination secondary to inflammation and increased vascular permeability following increased antigen load [26]. The result is a difference in conduction velocity among the axons. The motor unit potentials will be more dsynchronised with longer conduction distances, which may lead to less complete summation and altered phase cancellation leading to changes of the CMAP morphology. Demyelination occurs more frequently in patients during reaction episodes and it is characterized by TD, even in the group of patients without clinical signs of neuritis. This could be caused by silent neuritis.

The role of cytokines and inflammatory milieu in the establishment of demyelination was not fully understood. Here, we evaluated serum levels of both TNF and IL-6 in leprosy patients with or without CB. Our findings showed an increase of IL-6 in leprosy neuropathy patients with nerve CB in comparison with patients without CB (P = 0.002). The triggering mechanisms of reactional episodes are not yet agreed upon. However, when we evaluated the IL-6 levels in leprosy patients without reactional episodes or neuritis, we observed that concentrations of this cytokine were still significantly higher in patients with ulnar CB, suggesting that IL-6 may play a critical role in the specific stage of nerve damage related to temporal dispersion, despite circumstances with acute cytokine presence, such as reactional episodes. The results presented in this study open new perspectives to understand the clinical manifestations of the conduction block and the behavior of the immune system during the process of nerve impairment.

Clinical signs such as evaluation of sensory and motor impairment, nerve enlargement and bilateral ulnar nerve injury associated with eletrodiagnostic criteria such as demyelinating finds, specifically temporal dispersion, could be tools to help us decide on the best conduct in patients with elbow ulnar neuropathy, and more specifically, to decide if we will indicate a nerve biopsy to for the diagnosis of pure neural leprosy.

## Supporting information

**S1 File.**
(XLSX)

**S2 File.**
(XLSX)

## Author Contributions

**Conceptualization:** Márcia Jardim, Euzenir N. Sarno.

**Formal analysis:** Márcia Jardim, Robson T. Vital, Ximena Illarramendi, Mariana Hacker, Roberta O. Pinheiro.

**Funding acquisition:** Izabela J. R. Pitta.

**Investigation:** Márcia Jardim, Robson T. Vital, Beatriz Junqueira, Izabela J. R. Pitta, Roberta O. Pinheiro, Euzenir N. Sarno.

**Methodology:** Márcia Jardim, Robson T. Vital, Izabela J. R. Pitta, Roberta O. Pinheiro, Euzenir N. Sarno.

**Project administration:** Euzenir N. Sarno.

**Resources:** Beatriz Junqueira, Roberta O. Pinheiro.

**Software:** Ximena Illarramendi, Mariana Hacker.

**Supervision:** Euzenir N. Sarno.

**Writing – original draft:** Márcia Jardim.

**Writing – review & editing:** Ximena Illarramendi, Roberta O. Pinheiro, Euzenir N. Sarno.

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
