## [Decision Letter · Decision Letter 0]

6 Jul 2021

PONE-D-21-15552

The red flags of ulnar neuropathy in leprosy

PLOS ONE

Dear Dr. Jardim,

Thank you for submitting your manuscript to PLOS ONE. After careful consideration, we feel that it has merit but does not fully meet PLOS ONE’s publication criteria as it currently stands. Therefore, we invite you to submit a revised version of the manuscript that addresses the points raised during the review process.

We look forward to receiving your revised manuscript.

Kind regards,

Leila Harhaus

Academic Editor

PLOS ONE

Journal Requirements:

[Ministry of Health, Oswaldo Cruz Foundation]

 [The authors received no specific funding for this work.]

Reviewers' comments:

Reviewer's Responses to Questions

**Comments to the Author**

1. Is the manuscript technically sound, and do the data support the conclusions?

Reviewer #1: Partly

2. Has the statistical analysis been performed appropriately and rigorously? 

Reviewer #1: Yes

3. Have the authors made all data underlying the findings in their manuscript fully available?

Reviewer #1: Yes

4. Is the manuscript presented in an intelligible fashion and written in standard English?

Reviewer #1: Yes

5. Review Comments to the Author

Reviewer #1: Authors investigate clues to the diagnosis of pure neuritic leprosy.

The aims are important as these cases can be challenging.

Major concerns:

The investigative profile used is grossly inadequate. There is no information provided on the use of ultrasound and MRI neurography findings. Both of these have been studied in leprosy and some conclusions are already available.

Patients of lepra reactions perhaps should not be considered in this manuscript as the diagnosis is more clear during the reaction state and the electrophysiology findings are different in relation to the acutely altered immunity.

Minor comments:

Introduction could be shortened

6. PLOS authors have the option to publish the peer review history of their article (what does this mean?). If published, this will include your full peer review and any attached files.

Reviewer #1: **Yes: **Satish Khadilkar

---

## [Author Response · Author response to Decision Letter 0]

14 Oct 2021

Academic Editor:

1 - We perform the modifications to meet the style requirement according to: https://journals.plos.org/plosone/s/file?id=wjVg/PLOSOne_formatting_sample_main_body.pdf and https://journals.plos.org/plosone/s/file?id=ba62/PLOSOne_formatting_sample_title_authors_affiliations.pdf

2 – We include the tables as part of the main manuscript and we removed the individual files. 

3- We removed any funding-related text from the manuscript. We would like to keep the Funding Statement: The authors received no specific funding for this work.

Reviwer:

Reviwer: The investigative profile used is grossly inadequate. There is no information provided on the use of ultrasound and MRI neurography findings. Both of these have been studied in leprosy and some conclusions are already available.

Response: We are grateful for the possibility of discussing the importance of complementary methods in the diagnostic suspicion of the pure neural form of leprosy, including imaging. However, the gold standard for this diagnosis is the histopathological study, which requires an invasive and definitive procedure, a nerve biopsy. The aim of this study was to show that some neurophysiological findings may rise the suspicion of the diagnosis. This way, we will be able to increase the sensitivity and specificity of the indication of nerve biopsy for these patients. We agree that lately, studies evaluating the role of ultrasonography and the findings of magnetic resonance neurography have been available. However, they either describe specific findings in patients with pure neural leprosy. In some articles, the presence of Doppler was described in neuritis. It is important in patients with general leprosy, but pure neural leprosy patients have a lower incidence of acute neuritis, so US findings are less useful in this group of patients.

Reviewer : Patients of lepra reactions perhaps should not be considered in this manuscript as the diagnosis, is more clear during the reaction state and the electrophysiology findings are different in relation to the acutely altered immunity.

Response: Excellent comment, but patients with leprosy reactions were used precisely because the diagnosis of neuritis is more evident in this group of patients. Neurophysiological and clinical findings are well defined during reactional episodes. Patients with a pure neural leprosy do not have evident clinical findings of neuritis (pain, thickening and sensory dysfunction, associated or not with motor dysfunction), neurophysiological findings help in this diagnosis.

Rewier: Minor comments:

Introduction could be shortened

Response: According to the suggestion of the referee we short a little one the introduction (p. 3, lines 4,5 and 6).

---

## [Editor Report · Decision Letter 1]

27 Oct 2021

The red flags of ulnar neuropathy in leprosy

PONE-D-21-15552R1

Dear Dr. Jardim,

We’re pleased to inform you that your manuscript has been judged scientifically suitable for publication and will be formally accepted for publication once it meets all outstanding technical requirements.

Kind regards,

Leila Harhaus

Academic Editor

PLOS ONE

---

## [Editor Report · Acceptance letter]

4 Nov 2021

PONE-D-21-15552R1 

The red flags of ulnar neuropathy in leprosy 

Dear Dr. Jardim:

I'm pleased to inform you that your manuscript has been deemed suitable for publication in PLOS ONE. Congratulations! Your manuscript is now with our production department. 

Kind regards, 

on behalf of

Prof. Dr. med. Leila Harhaus 

Academic Editor

PLOS ONE